# Syntheses and Applications of 1,2,3-Triazole-Fused Pyrazines and Pyridazines

**DOI:** 10.3390/molecules27154681

**Published:** 2022-07-22

**Authors:** Gavin R. Hoffman, Allen M. Schoffstall

**Affiliations:** Department of Chemistry and Biochemistry, University of Colorado Colorado Springs, Colorado Springs, CO 80918, USA; ghoffman@uccs.edu

**Keywords:** synthesis, 1,2,3-triazole, fused 1,2,3-triazole, 1,2,3-triazolo[4,5-*b*]pyrazine, 1,2,3-triazolo[4,5-*c*]pyridazine, 1,2,3-triazolo[4,5-*d*]pyridazine, 1,2,3-triazolo[1,5-*a*]pyrazine, 1,2,3-triazolo[1,5-*b*]pyridazine, triazolopyrazine, triazolopyridazine, practical applications

## Abstract

Pyrazines and pyridazines fused to 1,2,3-triazoles comprise a set of heterocycles obtained through a variety of synthetic routes. Two typical modes of constructing these heterocyclic ring systems are cyclizing a heterocyclic diamine with a nitrite or reacting hydrazine hydrate with dicarbonyl 1,2,3-triazoles. Several unique methods are known, particularly for the synthesis of 1,2,3-triazolo[1,5-*a*]pyrazines and their benzo-fused quinoxaline and quinoxalinone-containing analogs. Recent applications detail the use of these heterocycles in medicinal chemistry (c-Met inhibition or GABA_A_ modulating activity) as fluorescent probes and as structural units of polymers.

## 1. Introduction

Within the 1,2,3-triazole-fused pyrazines and pyridazines, a series of congeners exists depending on whether a nitrogen atom occupies a position at the ring fusion (Figure 1).

We became interested in structures containing heterocyclic nuclei **2**, **4**, **6**, **8** and **10** following reports detailing potent mesenchymal–epithelial transition factor (c-Met) protein kinase inhibition, such as the current clinical candidate Savolitinib [1] (Figure 2, Structure A) and specifically those containing substructures **2** and **8** [1,2]. In addition to c-Met inhibition, structures containing these heterocyclic nuclei have shown GABA_A_ allosteric modulating activity [3] (Figure 2, Structure B), have been incorporated into polymers for use in solar cells [4,5] (Figure 2, Structure C), and have demonstrated β-secretase 1 (BACE-1) inhibition [6] (Figure 2, Structure D). Their piperazine derivatives have demonstrated potent PDP-IV inhibition [7].

Emphasis in this review is placed on the more common derivatives of **2** and **8**. In comparison to the heterocyclic scaffolds outlined in Figure 2, derivatives of **4**, **6** and **10** are less common in the literature. Among fused heterocycles containing the more well-known fused 1,2,4-triazoles, both 1,2,4-triazolo[1,5-*a*]pyrimidines [8] and 1,2,4-triazolo[4,3-*a*]pyrazines [9] have been recently reviewed. Kumar and coworkers [10] surveyed 1,2,3-triazoles fused to various rings, both aromatic and non-aromatic. In the present review, we address approaches to the synthesis of 1,2,3-triazole-fused pyrazines and pyridazines and their related congeners, while setting two limitations:This review covers synthetic methods of preparing structures containing fused heterocycles **2**, **4**, **6**, **8**, **10** (Figure 1). Tricyclic and tetracyclic congeners containing these heterocycles are included.1,2,3-Triazolopyrimidines do not appear in this review. They have received attention in the literature on purine chemistry [11,12,13].

1,2,3-Triazolopyrimidines, which form the core structure of 8-azapurines, 8-azaadenines, and 8-azaguanines, have been well-studied and reviewed [13,14,15] owing to their similarity to the respective nucleobases. With both scope and limitations in place, this review addresses synthetic approaches to the 1,2,3-triazolodiazine family: 1,2,3-triazolo[4,5-*b*]pyrazine, 1,2,3-triazolo[4,5-*c*]pyridazine, 1,2,3-triazolo[4,5-*d*]pyridazine, 1,2,3-triazolo[1,5-*a*]pyrazine, and 1,2,3-triazolo[1,5-*b*]pyridazine. The literature covered includes articles published since the most recent review of each type of compound, or earlier if no review exists. Reports are covered until the spring of 2022 and exclude tetrahydro-derivatives.

## 2. Synthetic Approaches

This overview of synthetic methods is organized according to the type of heterocycle. In the case of 1*H*-1,2,3-triazolo[1,5-*a*]pyrazines, methods are subdivided into pyrazines and benzopyrazines. Reaction times are included along with solvents, catalysts, and other reagents in most examples. Commercial availability of precursors is emphasized where applicable.

### 2.1. Syntheses of 1H-1,2,3-Triazolo[4,5-b]pyrazines

One of the first reported preparations of a 1*H*-1,2,3-triazolo[4,5-*b*]pyrazine came from Lovelette and coworkers [16], who utilized condensation of a 4,5-diamino-1,2,3-triazole, **14**, and a 1,2-dicarbonyl compound **15** (Figure 1) to give the desired triazolopyrazines **16** in yields in the range 30–35%. A useful precursor, 4,5-diamino-1,2,3-1*H*-triazole **14**, was prepared by reacting carbamate **13** with a strong base. This carbamate was readily prepared from the carbonyl azide by refluxing in ethanol. The carbonyl azide can be prepared from benzyl azide **11**, ethyl cyanoacetate **12**, and sodium ethoxide, all commercially available starting materials.

Dicarbonyl compounds included glyoxal (R_1_ = R_2_ = H), benzil (R_1_ = R_2_ = Ph), and others. This was one of the first reports of 1,2,3-triazole-fused pyrazines, highlighted within a study of fused 1,2,3-triazoles. This method offers three-point diversity, one from the triazole substituent, and the other two from the respective dicarbonyl substituents. Despite this, a potential drawback lay in the restriction to a symmetrically substituted 1,2-dicarbonyl species to avoid mixtures of isomers. Indeed, the authors noted the two condensation products using an asymmetrically substituted diketone, where R_1_ = CH_3_ and R_2_ = H, as being indistinguishable.

Monge and coworkers [17] prepared benzo-fused 1*H*-1,2,3-triazolo[4,5-*b*]pyrazines through the acid-catalyzed cyclization of 2-azido-3-cyanoquinoxaline, **18**, obtained from 2-chloro-3-cyanoquinoxaline **17**, yielding 1-hydroxy-1*H*-1,2,3-triazolo[4,5-*b*]quinoxaline **19** (Figure 2) in 52% yield. Though uncommon, acid-catalyzed cyclization of *ortho*-substituted azidocyanoaryl species may represent an underutilized method of obtaining structures with the 1,2,3-triazolo[4,5-*b*]pyrazine core. Despite this, the use of costly starting materials hinders wider applicability.

Unexpectedly, Starchenkov and coworkers [18] determined that, upon treatment of diamine **20** with trifluoroacetic anhydride (TFAA) and HNO_3_ and proceeding via intermediate **21**, triazolopyrazine N-oxide **22** was formed (Figure 3). This was one of the first reports of the preparation of a fused 1,2,3-triazole 2-N-oxide, namely [1,2,5]oxadiazolo[3,4-*b*][1,2,3]triazolo[4,5-*e*]pyrazine-6-oxide **22**, formed in 92% yield.

Forming a mesoionic ring system while studying luminescence, Slepukhin and coworkers [19] obtained the 1*H*-1,2,3-triazolo[4,5-*b*]pyrazine core within the azapentalene inner salt **27** in 50% yield after intramolecular cyclization of 8-(benzotriazole-1-yl)tetrazolo[1,5-*a*]pyrazine **25** in refluxing DMF, causing loss of nitrogen via intermediate **26** and formation of 5*H*-pyrazino[2′,3′:4,5][1,2,3]triazol[1,2-*a*]benzotriazol-6-ium, inner salt **27** (Figure 4). Pyrazine **25** was prepared in 48% yield by nucleophilic aromatic substitution of chloride by the benzotriazolyl ion after deprotonation of 1*H*-1,2,3-benzotriazole **24** by carbonate.

Azapentalenes, containing the 1*H*-1,2,3-triazolo[4,5-*b*]pyrazine nucleus, have gained attention for their useful properties, such as in luminescence and complexation [19]. Compounds of this type have demonstrated low toxicity, high solubility, and other properties desirable as potential fluorescence probes [20]. This intramolecular approach has remained popular in obtaining various substituted azapentalenes, another example being that of Nyffenegger and coworkers [21]. Here, the azapentalene, 5*H*-pyrazolo[1′,2′:1,2][1,2,3]triazolo[4,5-*b*]pyrazin-6-ium, inner salt, **31**, was obtained in yields up to 85% via cyclization with loss of nitrogen after amination of 2-azido-3-chloropyrazine, **28**, with either pyrazole **29** or 1,2,4-triazole affording 2-azido-3-(1*H*-pyrazol-1-yl)pyrazine **30** (Figure 5). Other derivatives using nitro-substituted pyrazoles were formed in yields in the range 63–97%. This method offers convenience in that a precursor to **28**, 2,3-dichloropyrazine, is commercially available.

Notably, in addition to having an azido group substituted *ortho* to the pyrazole of **30** [20,21], reports have also made use of the respective amine via ring closure by displacement of an N-iodonium intermediate by an adjacent nitrogen atom of the attached pyrazole to form azapentalenes [22,23]. Compounds of this type have been thoroughly characterized via NMR spectroscopy [24]. A Pfizer patent [25] filed in 2007 detailed the use of either isoamyl nitrite in DMF or NaNO_2_ in aqueous acetic acid, after first aminating commercially available 2-amino-3,5-dibromopyrazine **32** in the presence of a sterically hindered base, *N,N*-diisopropylethylamine (DIPEA), then treating diaminopyrazine **33** with nitrite to form 3,5-disubstituted 1*H*-1,2,3-triazolo[4,5-*b*]pyrazine **34** (Figure 6).

The use of nitrite for triazole cyclization, via the nitrosonium ion, has also been reported by Ye and coworkers [26,27], Cui and coworkers [2] (who were cited in the original patent [25]), and others. Thottempudi and coworkers [28] used a combination of TFAA/HNO_3_ as an in situ nitronium source, giving a triazole 2-N-oxide, while Jia and coworkers [1], and others [29,30] used nitrosonium generated from nitrite. Both syntheses offer straightforward introduction of the triazole based on the amine chosen during amination. They also have the advantage of short reaction times and little or no required purification. Likely owing to these benefits, cyclization using nitrite to generate nitrosonium ion, such as in **33** to **34** (Figure 6), continues to dominate reports in the literature. Indeed, the reaction of various diazinyl diamines with nitrite represents a central theme throughout the discussion of syntheses of 1*H*-1,2,3-triazole-fused pyrazines and pyridazines.

### 2.2. Syntheses of 1,2,3-Triazolo[1,5-a]pyrazines

More well-known than 1,2,3-triazolo[4,5-*b*]pyrazines are the fused [1,5-*a*]pyrazine derivatives. While benzo[*b*]pyrazines (i.e., quinoxalines) are not commonly encountered as part of 1,2,3-triazolo[4,5-*b*]pyrazines, they are widespread in the literature in compounds containing the 1,2,3-triazolo[1,5-*a*]pyrazine nucleus. Therefore, this section is organized into the syntheses of benzo-fused structures (e.g., 1,2,3-triazolo[1,5-*a*]pyrazines containing quinoxaline or quinoxalinone), and those that are bicyclic 1,2,3-triazolo[1,5-*a*]pyrazines. A recent brief review of 4,5,6,7-tetrahydro[1,2,3]triazolo[1,5-*a*]pyrazines has been published [31]. An earlier review detailed aspects of the chemistry of 1,2,3-triazolo[1,5-*a*]pyrazines [32]. A review on the synthesis of triazoloquinazolines also appeared in 2016 [33].

#### 2.2.1. Syntheses of Bicyclic 1,2,3-Triazolo[1,5-a]pyrazines

The first method of synthesizing a 1,2,3-triazolo[1,5-*a*]pyrazine by Wentrup [34] was, at the time, the synthesis of a novel purine isomer. Wentrup utilized the thermolysis of 5-(2-pyrazinyl)tetrazole **36** (400 °C, 10^−5^ Torr), affording **38**, 1,2,3-triazolo[1,5-*a*]pyrazine in 20% yield proceeding via diazo intermediate **37**. The precursor 2-(2*H*-tetrazol-5-yl)pyrazine **36** was readily prepared from 2-cyanopyrazine, **35**, upon treatment with hydrazoic acid generated in situ from ammonium chloride and sodium azide (Figure 7). This method, while suffering from harsh reaction conditions and poor yields, was the first utilizing intramolecular cyclization of diazo intermediates in the formation of 1,2,3-triazolo[1,5-*a*]pyrazines. Lead tetraacetate oxidation of the hydrazone of pyrazine-2-carbaldehyde similarly gave **38** in 75% yield [35].

In addition to syntheses of neutral compounds of this type, several reports have appeared for the preparation of fused pyrazinium salts. A method by Beres and coworkers [36] afforded 1-(4-bromophenyl)-3-methyl-1,2,3-triazolo[1,5-*a*]pyrazinium tetrafluoroborates **31** in 55% yield (when R_1_ = *p*-chlorophenyl) and 81% yield (when R_1_ = CH_3_) after reaction of 4-bromophenylhydrazones **39** (prepared from the respective 2-pyrazinyl ketone) with tribromophenol bromine (TBB) and NH_4_BF_4_ (Figure 8). When R_1_ = CH_3_, the yield of **40** was 81%. Interestingly, after treatment of **40** with pyrrolidine in methanol, the ring-opened 2-aza-1,3-butadienes can be valuable starting materials for other conversions. For example, a ring-opened triazolyl-2-aza-1,3-butadiene was converted to a fused pyridine after treatment with N-phenylmaleinimide, or an imidazoline when treated with tosyl azide [36].

Methods have been reported for the preparation of 1,2,3-triazolo[1,5-*a*]pyrazinones. In work by Nein and coworkers [37,38], the reaction of 5-hydroxy-*N*-diphenyl-1*H*-1,2,3-triazole-4-carboxamide **41** with chloroacetonitrile in DMF and base gave the alkylated product **42**, which, after refluxing in sodium ethoxide, gave 6-amino-4-oxo-2,5-diphenyl-4,5-dihydro-2*H*-1,2,3-triazolo[1,5-*a*]pyrazinium-5-olate **43** in 80% yield (Figure 9). They proposed the geometry of 3-phenacyl- and 3-cyanomethyl derivatives of triazolium-5-olates indicated interaction of the carboxamide nitrogen at position 4 of the triazole with cyano groups, which was then confirmed experimentally after obtaining the desired mesoionic **42** [37].

Similarly involving reaction of triazolium olates such as **42**, during a synthesis of 1,2,5-triazepines by Savel’eva and coworkers [39], [1,5-*a*]triazolopyrazines were formed as byproducts (5–7%) from the intramolecular cyclization of 1-amino-3-(*p*-phenacyl)-4-{[2-(1-methylethylidene)hydrazino]carbonyl}-[1,2,3]-triazolium-5-olates. Jug and coworkers [40] took a novel approach for the reaction of 4-(ethoxymethylene)-2-phenyloxazol-5(4*H*)-one **44** with commercially available diaminomaleonitrile **45**, forming adduct **46** which, after conversion to triazole **47** with nitrite, afforded the substituted 1,2,3-triazolo[1,5-*a*]pyrazine **48** (Figure 10). Later, derivatives of **48** such as ethyl 4-amino-3-cyano-1,2,3-triazolo[1,5-*a*]pyrazine-6-carboxylate were further reacted by Trcek and coworkers [41] to form 1,2,3-triazolo[1,5-*a*]-1,2,4-triazolo[5,1-*c*]pyrazines in 55–65% yield.

Raghavendra and coworkers [42] reported a triazolopyrazine synthesis employing solid-phase polystyrene *p*-toluenesulfonyl hydrazide, a common carbonyl scavenging resin. After reaction of the polystyrene *p*-toluenesulfonyl hydrazide **49** with an acetylpyrazine **50** in the presence of 5% TiCl_4_ in MeOH, hydrazone **51** was obtained. Reaction of **51** with morpholine gave the desired 1,2,3-triazolo[1,5-*a*]pyrazines, **52** (Figure 11), in yields ranging from 33–62%. This regiospecific, traceless protocol represented the first solid-phase assisted synthesis of a triazolopyrazine and was also used for the synthesis of several non-fused 1,2,3-triazoles in the same report in yields up to 60%.

Copper-catalyzed [3 + 2] cycloaddition of propiolamide **53**, followed by halide displacement to form a fused product, was utilized in the synthesis of saturated derivatives of 1,2,3-triazolo[1,5-*a*]pyrazine (i.e., triazolopiperazines) **54** in 80% yield [43] (Figure 12). Koguchi and coworkers used ynones and β-amino azides to afford 6,7-dihydro-1,2,3-triazolo[1,5-*a*]pyrazines. These authors verified that the one-pot reaction gave cycloaddition of the alkyne and azide first, followed by reaction of the amine with the ketone [44].

#### 2.2.2. Syntheses of Benzo-Fused 1,2,3-Triazolo[1,5-a]pyrazines

One of the first reported preparations of a 1,2,3-triazolo[1,5-*a*]pyrazine by Kauer and coworkers [45] started with dimethyl l-(*o*-nitrophenyl)-l*H*-triazole-4,5-dicarboxylate **59**, and upon treatment with tributyl phosphine in refluxing toluene, afforded methyl 4-methoxy-1,2,3-triazolo[3,4-*a*]quinoxaline-3-carboxylate **60** (Figure 13). Triazole **59** was readily prepared from *o*-azidonitrobenzene **57** (which in turn was prepared from *o*-chloronitrobenzene **55** or *o*-aminonitrobenzene **56**) and dimethyl acetylenedicarboxylate **58** in CHCl_3_.

Through a different approach, Cue and coworkers [46] accessed 1,2,3-triazolo[l,5-*a*]quinoxaline N-oxides **62** in yields ranging from 52–70% by cyclization of quinoxaline-3-carboxaldehyde-1-oxide-*p*-toluenesulfonylhydrazone **61** (Figure 14). The starting sulfonylhydrazone **61** was prepared by reaction of a 3-substituted quinoxaline N-oxide with *p*-toluenesulfonylhydrazide [45].

For the intramolecular cyclization of *ortho*-substituted amines to prepare 1,2,3-triazoles using nitrite, as is commonly reported for non-fused derivatives [2,25], Ager and coworkers [47] illustrated that the amines used in cyclization do not need to be primary. Through reaction of a secondary amine within a ring and a primary amine **63** with isoamyl nitrite in chloroacetic acid, they obtained 1,2,3-triazoles fused to both lactones and lactams, **64**, in yields in the range 54–76% (Figure 15). In the case of lactams, 1,2,3-triazoloquinoxalinones were formed.

Synthesizing compounds of the same type, Bertelli, and coworkers [48] first formed a triazole diester on a ring *ortho* to a nitro group, **65**, which was intramolecularly cyclized to form ethyl 4,5-dihydro-4-oxo-[1,2,3]triazolo[1,5-*a*]quinoxaline-3-carboxylate, **66** (Figure 16). This reaction was conducted by hydrogenation with a 10% Pd/C catalyst or by reaction with FeCl_3_ and Fe powder. Biagi and coworkers [49] cyclized the triazole diester into 1,2,3-triazoloquinoxalinone **66** with 10% Pd/C in ethanol in an excellent 98% yield. Shen and coworkers further modified the ester group of **66** to prepare a derivative suitable for biological testing [50].

Abbott and coworkers [51] prepared 1,2,3-triazoloquinoxalines in an analogous manner, but opted for use of an amide instead of a nitro group, giving mesoionic 1,2,3-triazoles **68**, which were derived from the lithium salt of [2-(acetylamino)phenyl]amino acetic acid **67** (Figure 17). A series of 1,2,3-triazoloquinoxalines, **69**, was synthesized after cyclization with *p*-toluenesulfonic acid (*p*-TSA) in refluxing toluene in yields in the range 16–59%.

Saha and coworkers [52] used the intramolecular cyclization of *ortho*-substituted anilines with tethered 1,2,3-triazoles, **72**, a Pictet–Spengler reaction, to form 1,2,3-triazoloquinoxalines **73** in yields in the range 61–70% (Figure 18). This sequence offers two-point diversity: one from **72**, and the other from an aryl aldehyde **73**. The prerequisite triazole **72** was conveniently prepared from readily available starting materials, including *o*-fluoronitrobenzene **70**, phenylacetylene **71**, and sodium azide.

Chen and coworkers [53] used a novel approach for the synthesis of 4-(trifluoromethyl)-1,2,3-triazolo[1,5-*a*]quinoxaline **76** via cascade reactions of N-(*o*-haloaryl)alkynylimine **75** with sodium azide in the presence of copper iodide and L-proline (Figure 19). Among a series of amine-containing catalysts, L-proline resulted in a 98% isolated yield, while tetramethylethylenediamine and *N*,*N′*-dimethylethylenediamine gave lower yields, and higher percentages of the uncyclized imine product.

Using photoredox catalysis, He and coworkers [54] used [*fac*-Ir(ppy)_3_] as a photocatalyst to afford the corresponding 1,2,3-triazoloquinoxaline **78** from isonitrile **77** in 60% yield (Figure 20). Due to poor solubility of the catalyst, ACN resulted in decreased yields compared to DMF. This work is a rare example of free-radical generation of 1,2,3-triazole-fused ring systems, as cyclohexyl radicals are proposed to have formed from phenyliodine(III)dicarboxylate. The radicals yield isonitrile carbon radicals, followed by reaction with carbon 5 of the triazole. Various fused rings were synthesized in addition to 1,2,3-triazoles including tetrazoles, pyrazoles, and imidazoles in yields as high as 80%.

In the presence of Cu(OAc)_2_ and base in DMSO/THF, Li and coworkers [55] reported an efficient one-pot synthesis of 1,2,3-triazolo[1,5-*a*]quinoxalines **81** from 1-azido-2-isocyanoarenes **79** in yields in the range 40–84% (Figure 21). They outlined the option of using terminal acetylenes **80** or substituted acetaldehydes **82**, the former being cyclized into **81** in one step (in yields ranging from 40–83%), and the latter forming uncyclized triazole **83,** which was annulated using Togni’s reagent II and tetra-*n*-butylammonium iodide (TBAI), forming **84**, or phenylboronic acid, forming **85**. Derivatives of **84** were prepared in yields in the range 26–78%, and one synthesis of **85** yielded 86%.

Owing to the versatility of intermediate **83**, many functionalized 1,2,3-triazoloquinoxalines were prepared, and indeed, Li and coworkers reported several compounds containing the 1,2,3-triazolo[1,5-*a*]quinoxaline core with a variety of functionalities. Additionally, in this report, the fused products were further reacted into diversified quinoxaline derivatives via Rh(II)-catalyzed carbenoid insertion reactions [55].

Employing a Pd-catalyzed intramolecular cyclization of triazole **86**, Kotovshchikov and coworkers [56] synthesized 3-butyl-[1,2,3]triazolo[1,5-*a*]quinoxalin-4(5*H*)-one **87** in 77% yield. As this reaction was conducted under CO (1 atm), the carbonyl carbon of the quinoxalinone was introduced by Pd-catalyzed insertion of CO (Figure 22).

Xiao and coworkers [43] and Chen and coworkers [57] used in situ conversion of N-propargyl-N-(2-iodoaryl)amides **88** to azides, which underwent 1,3-dipolar cycloaddition with the adjacent alkyne to form substituted 1,2,3-triazolo[1,5-*a*]quinoxalines **89** (Figure 23) in yields in the range 58–91%. Chen and coworkers suggested that cycloaddition might occur first. The sequence was conducted in the presence of DIPEA and 1,2-dimethylethylenediamine (DMEDA).

Preparative thermolysis of tetrazoloquinoxaline **90** proceeded by loss of nitrogen through diazo intermediate **91** and then to 1,2,3-triazolo[1,5-*a*]quinoxaline **92** in 67% yield (Figure 24) [58]. Using a ring-closure method similar to that used by both Raghavendra and coworkers [42] and Cue and coworkers [46], Vogel and Lippmann [59] developed a route to derivatives of **92** in 47–89% yield via conversion from tosylhydrazones **93** using base (Bamford-Stevens conditions) or, in certain cases, heat (Figure 24).

Overall, there exist diverse methods for the synthesis of both bicyclic 1,2,3-triazolo[1,5-*a*]pyrazines and 1,2,3-triazolo[1,5-*a*]quinoxalines.

### 2.3. Syntheses of 1H-1,2,3-Triazolo[4,5-d]pyridazines

Livi and coworkers [60] reviewed syntheses of this heterocyclic system covering reports prior to 1996. Another review on condensed 1,2,3-triazoles appeared in 2008, which includes synthesis of 1*H*-1,2,3-triazolo[4,5-*d*]pyridazines [32]. Here, we summarize both older and newer reports. A common theme in the literature regarding the synthesis of 1*H-*1,2,3-triazolo[4,5-*d*]pyridazines is the reaction of 1,2,3-triazole dicarbonyl species with hydrazine hydrate. This yields a diacylhydrazide, which can be cyclized with either high heat or acid. One of the first examples (Figure 25) is from Fournier and Miller [61], who used 2-(4,5-dibenzoyl-1*H*-1,2,3-triazol-1-ylmethyl)-3,4,6-trimethylhydroquinone diacetate and hydrazine hydrate in ethanol to form 4,5-diphenyl-1*H-*1,2,3-triazolo[4,5-*d*]pyridazine. In a comparable manner, Erichomovitch [62] used triazole diesters **94** to obtain diacylhydrazides **95**, which were heated to form 1*H-*1,2,3-triazolo[4,5-*d*]pyridazines **96** in 80% yield with loss of hydrazine.

Janietz and coworkers [63] developed a scheme that proceeded through dichlorotriazole **97**, which, after conversion to a dinitrone and subsequent treatment with acid, afforded the dialdehyde **98**, which cyclized to form the desired 1*H*-1,2,3-triazolo[4,5-*d*]pyridazine **99** after treatment with hydrazine (Figure 26).

Reports of forming 1,2,3-triazolo[4,5-*d*]pyridazones or pyridazines using this method include those of Gilchrist [64,65], Milhelcic [66], Ramesh [67], Theocharis [68], Bussolari [69], Biagi [70,71,72], Abu-Orabi [73], Ramanaiah [74], Bankowska [75], and others [5,76,77,78].

Martin and Castle [79] used ring closure by nitrosonium ion in their treatment of a 4,5-diamino-6-pyridazinone **101** in forming 3,5-dihydro-4*H-*1,2,3-triazolo[4,5-*d*]pyridazin-4-one **102** in 91% yield (Figure 27). Commercially available 4,5-dichloro-3(2*H*)-pyridazinone **100** was converted to **101** in three steps. Similar methods of reacting substituted diaminopyridazines with nitrite have been conducted by Yanai [80] (conversion of **103** to **104** in Figure 27), Chen [81], Draper [82], and Mataka [83].

Smolyar and coworkers [84] reported a novel synthesis of a 1*H-*1,2,3-triazolo[4,5-*d*]pyridaz-4-one, **106** by a ring-opening/ring-closing “cyclotransformation” involving treatment of 1*H-*1,2,3-triazole-fused 5-nitropyridin-2(1*H*)-ones **105** with a large excess of hydrazine hydrate (Figure 28). They reported that after heating for 3–4 h, at 140 °C, the desired pyrazinone was obtained in 86% yield with no chromatography required. 5-Nitropyridin-2(1*H*)-ones fused with benzene and pyridine were also studied in this report.

A number of methods exist for the preparation of molecules containing the 1,2,3-triazolo[4,5-*d*]pyridazine core, the majority of which involve the treatment of 1,2,3-triazole dicarbonyl species with hydrazine hydrate followed by acid or heat-promoted cyclization, or the cyclization of a diaminopyridazine with nitrite.

### 2.4. Syntheses of 1,2,3-Triazolo[1,5-b]pyridazines

Despite being reported as early as 1949 by Schofield and coworkers [85] in their study of cinnolines, 1,2,3-triazolo[1,5-*b*]pyridazines remain rare in the literature, in part owing to few methods available for their synthesis. While synthesizing azepinones, Evans and coworkers [86] instead serendipitously obtained 3,6-diphenyl-1,2,3-triazolo[1,5-*b*]pyridazine **108**. This was obtained from the intramolecular cyclization of diketo-oxime **107** (Figure 29) after refluxing in HCl. This gave up to 22% of a pyrazinylhydrazone byproduct. A similar method in the same report used HOAc, but this resulted in poor yields (about 15%) and up to three products.

A fluoroborate salt was prepared by Riedl and coworkers [87] in a manner similar to that of Beres and coworkers [36]. The acyl-substituted pyridazine, **111**, after treatment with *p*-bromophenyl hydrazine hydrochloride **112** gave the hydrazone **113**. Tribromophenol bromine (TBP) in DCM afforded the desired ring-closed product **114** in 67% yield (Figure 30). The initial bromide salt was converted to the fluoroborate salt with 40% fluoroboric acid in ACN. Ketone **111** was prepared by the same group via reaction of a commercially available 3-cyanopyridizine **109** with *p*-chlorophenylmagnesium bromide **110**, also synthesized from commercially available *p*-chlorobromobenzene and Mg. This was followed by acidic workup to afford the desired ketone. Compounds of this type were also prepared by Vasko and coworkers [88] using a similar method, which gave a 27% yield. A third method for the synthesis of 1,2,3-triazolo[1,5-*b*]pyridazines consisted of intramolecular oxidative ring closure of a hydrazone derived from **111** to afford the neutral 1,2,3-triazolo[1,5-*b*]pyridazine **115** [89]. Kvaskoff and coworkers employed MnO_2_ as an oxidant using a similar procedure [35,89,90], where purification by sublimation afforded the desired product **115** (where R_1_ = R_2_ = H) in 71% yield.

### 2.5. Syntheses of 1,2,3-Triazolo[4,5-c]pyridazines

More prevalent in the literature than 1,2,3-triazolo[1,5-*b*]pyridazines but still uncommon are the 1,2,3-triazolo[4,5-*c*]pyridazines. One of the first reports of such a compound came from Gerhardt and coworkers [91], whereas in previous reports, nitrite was used to cyclize 5-chloro-3,4-diaminopyridazine **116** to afford 7-chloro-3*H*-1,2,3-triazolo[4,5-*c*]pyridazine **117** (Figure 31) in 83% yield. Nitrite in the presence of an acid catalyst has been used for the synthesis of this heterocyclic ring system from the respective diaminopyridazines in other reports by Murakami [92], Lunt [93], Ramanaiah [74], and Owen [3].

In a report by Pokhodylo and coworkers [94], nitrite was used in the synthesis of a substituted 1,2,3-triazolo[4,5-*c*]pyridazine despite only having one amine group present (as opposed to other cyclizations, which have two amine groups present). For example, 4-(3,4-dimethoxyphenyl)-1-phenyl-1*H*-1,2,3-triazol-5-amine **118** was reacted with sodium nitrite and glacial acetic acid to give the desired 3-(4-chlorophenyl)-7,8-dimethoxy-3*H*-[1,2,3]triazolo[4,5-*c*]cinnoline **119** in 35% yield (Figure 32). Yields may have been low compared to other nitrite cyclizations due to the formation of a C-N bond directly with a carbon of an aromatic ring.

Daniel and coworkers [22] formed tricyclic ylides **121** in 65% yield by oxidative cyclization of the respective *ortho*-substituted amino pyridazine **120** (Figure 33). Unfortunately, compounds containing the 1,2,3-triazolo[4,5-*c*]pyridazine nucleus remain rare in the literature, and little is known of their biological or pharmacological properties.

## 3. Applications

Recent applications of the aforementioned heterocyclic systems, covering both medicinal and non-medicinal topics, are discussed in the following section.

### 3.1. Applications of 1H-1,2,3-Triazolo[4,5-b]pyrazines

In the last decade, 1*H*-1,2,3-triazolo[4,5-*b*]pyrazines have garnered an interest within the field of medicinal chemistry for serving as the scaffold of selective c-Met inhibitors. Medicinal studies of 1*H*-1,2,3-triazolo[4,5-*b*]pyrazines have extended well into the patent literature, with one patent even exploring antiviral efficacy against SARS-CoV-2 [95]. The first notable report of physiological activity came from Cui and coworkers [2], who reported the discovery of PF-04217903, a 1,2,3-triazolo[4,5-*b*]pyrazine that demonstrated potent (IC_50_ = 0.005 µM) and selective inhibition of over 200 c-Met kinases [2]. This heterocyclic scaffold in general gave rise to derivatives (altering substituents at the 2 and 6 ring positions) with potent inhibition, of which PF-04217903 was the best. This compound was selected as a preclinical candidate for the treatment of cancer [96].

Later, using PF-04217903 as a reference, Jia, and coworkers [1] reported the discovery of a compound now known as Savolitinib (Figure 3). This compound, also an exquisite c-Met inhibitor with an equal IC_50_ of 0.005 µM, demonstrated favorable pharmacokinetic properties in mice [1]. Savolitinib possessed equal potency. Having recently passed phase II clinical trials for the treatment of metastatic non-small cell lung cancer, papillary and clear cell renal cell carcinoma, gastric cancer, and colorectal cancer, Savolitinib has been granted conditional approval for use in China at the time of this review [97]. A review of c-Met inhibitors in non-small cell lung cancer has recently appeared [98].

Sirbu and coworkers [20] recently reported a novel class of small molecules containing the 1,2,3-triazolo[4,5-*b*]pyrazine scaffold with excellent properties for use as versatile fluorescent probes in optical imaging (Figure 4). Specifically, a phenyl ester derivative was used to dye HeLa cells in epifluorescence microscopy. Compared to commercially available LysoTracker Green DND-26, the tested triazolopyrazine derivative demonstrated comparable properties. In addition, it showed low cytotoxicity when evaluated in Alamar Blue assay (>95% cell viability up to 170 µM) and showed high solubility with a variety of desirable characteristics. A phenyl ester derivative, when evaluated as a dye in HeLa cells, showed high photostability and low cytotoxicity [20].

Intriguingly, another application lay in the monitoring of hypoxic regions within tumor cells. This was explored by Janczy-Cempa and coworkers [23], who looked at the fluorescent products produced after reduction of nitrotriazolopyrazine probes by nitroreductases (enzymes often overexpressed in tumor regions). Both probes studied (Figure 5) had very weak fluorescence in normoxic regions, but their reduction by nitroreductases led to a 15-fold increase in intensity in hypoxic regions. This was evaluated using the human melanoma cell line A2058. In contrast to the fluorescence probes developed by Sirbu and coworkers [20], probes in this study had substitutions on the pyrazine ring as opposed to the triazole-fused pyrazole. While additional work is still to be done, this report demonstrates the potential for these highly conjugated compounds to be useful in biomedical monitoring. Legentil and coworkers [99] obtained compounds similar to the structure on the right in Figure 5 in yields as high as 79%, which were used to develop a luminescence layered double-hydroxide filter. This material was dispersed into a polymer for use as a dye.

Overall, applications of compounds containing 1,2,3-triazolo[4,5-*b*]pyrazines in the current literature are focused on c-Met inhibition (i.e., the treatment of distinct types of cancers), and optical and/or cellular imaging, with triazapentalene-type molecules demonstrating a wide range of favorable characteristics as fluorescent probes.

### 3.2. Applications of 1H-1,2,3-Triazolo[4,5-c]pyridazines

After being initially evaluated by Gerhardt and coworkers [91] as potential purine antagonists, 1*H*-1,2,3-triazolo[4,5-*c*]pyridazines have since found broader interest within medicinal chemistry. In a report by Owen and coworkers [3], a 1*H*-1,2,3-triazolo[4,5-*c*]pyridazine was found to have GABA_A_ modulating activity during a structure–activity relationship study of the respective imidazolopyridazine. Compounds containing the 1,2,3-triazolo[4,5-*c*]pyridazine scaffold have been investigated in the patent literature for the treatment of Huntington’s disease [100] and as modulators of Janus-family kinase-related diseases [101].

Other recent patents have been filed regarding fused pyridazines with herbicidal activity, of which 1,2,3-triazolo[4,5-*c*]pyridazine is included [102]. In another recent patent, compounds of this type were implicated in controlling unwanted plant growth [103].

Reports of compounds containing the 1,2,3-triazolo[4,5-*c*]pyridazine scaffold are uncommon in the current literature beyond synthetic reports and patents. Undoubtedly, there is still work to be done in exploring the potential applications of this unique heterocyclic system.

### 3.3. Applications of 1H-1,2,3-Triazolo[4,5-d]pyridazines

In a recent development, Li, and coworkers [4] outlined a series of triazole-based structures for the construction of conjugated polymers for solar cells. In addition to demonstrating desirable properties as units incorporated into polymers (Figure 6), their reported synthetic route uses affordable, commercially available starting materials and produces units compatible with other monomers. Structures containing 1,2,3-triazolo[4,5-*d*]pyridazine components offer a privileged, conjugated unit for the construction of polymers owing in part to the convenient para substitution of the pyridazine ring and perpendicular N2 substitution of the triazole ring.

Another notable outcome of the study of 1,2,3-triazolo[4,5-*d*]pyridazines was that from Biagi and coworkers [104], who reported compounds of this type with high selectivity for the A_1_ receptor subtype in radioligand binding assays at bovine brain adenosine A_1_ and A_2A_ receptors. The most potent compound contained a 4-amino-substituted 7-hydroxy-1,2,3-triazolo[4,5-*d*]pyridazine, and after substitution of the hydroxyl group for a chlorine, affinity decreased and suggested a hydrogen-bond donating substituent at position 7 was critical for binding affinity.

### 3.4. Applications of 1,2,3-Triazolo[1,5-a]pyrazines

Among applications of compounds containing the 1,2,3-triazolo[1,5-*a*]pyrazine unit are those of benzo-fused 1,2,3-triazoloquinoxalines and saturated 1,2,3-triazole-fused piperidines. In a recent report by Pérez Morales and coworkers [105], a 1,2,3-triazoloquinoxalinone (Structure A, Figure 7) was identified via high-throughput screening as inducing expression of Rgg2/3-regulated genes in the presence of short hydrophobic pheromones at low concentrations. This work stemmed from interest in the Rgg2/3 quorum sensing circuit of the pathogen *Streptococcus pyogenes*, with the objective of manipulating and inhibiting the bacteria. After analyzing its mode of action, it was determined this compound directly uncompetitively inhibited recombinant PepO in vitro, and induced quorum sensing signaling by stabilizing short hydrophobic pheromones.

Based on the antidiabetic 1,2,4-triazolopiperazine-containing drug Sitagliptin (brand name Januvia), Shan and coworkers [7] identified a dipeptidyl peptidase (DPP) IV inhibitor containing a 1,2,3-triazolopiperazine (Structure B, Figure 7) for use in the treatment of type II diabetes.

Partially saturated 1,2,3-triazolo[1,5-*a*]pyrazines have demonstrated BACE-1 inhibition, an enzyme implicated in the formation of amyloid beta in Alzheimer’s disease. Oehlrich and coworkers [6] identified one such candidate, (*R*)-*N*-(3-(4-amino-6-methyl-6,7-dihydro-[1,2,3]triazolo[1,5-*a*]pyrazin-6-yl)-4-fluorophenyl)-5-cyanopicolinamide, (Structure C, Figure 7). This demonstrated an inhibition of the BACE-1 enzyme of pIC_50_ = 8.70.

These reports, while not exhaustive, demonstrate recent applications of compounds containing the 1,2,3-triazolo[1,5-*a*]pyrazine scaffold or congeners thereof. Particularly prominent in the literature are benzo-fused and piperazine-containing analogs.

### 3.5. Applications of 1,2,3-Triazolo[1,5-b]pyridazines

There are no reported applications of compounds containing the 1,2,3-triazolo[1,5-*b*]pyridazine ring system, and little regarding its physiological and/or pharmacological effects are known. Aside from one recent patent [106] regarding immunoregulatory functions, additional applications remain scarce at the time of this review.

## 4. Conclusions

In reviewing synthetic approaches to and reported applications of members of the 1,2,3-triazolodiazine family of fused bicyclic heterocycles, the following conclusions can be drawn regarding the most common synthetic methods and applications in the present literature:(a)*1,2,3-Triazolo[4,5-b]pyrazines*: The most common synthetic method is cyclization of an *ortho*-substituted diaminopyrazine [2], in which one of the amines does not need to be primary [18,25]. Given the current commercial availability and affordability of 2-amino-3,5-dibromopyrazine, this serves as a convenient starting material. Other methods include condensation of a dicarbonyl species with a 4,5-diamino-1,2,3-triazole [16], cyclization of a 2-azido-3-cyanoquinoxaline [17], or formation of azapentalenes from tetrazolopyrazines [19] or pyrazolopyrazines [21] with loss of nitrogen.*Primary Applications*: Primarily c-Met inhibition [1,2,25,96] and use as fluorescent probes in optical and/or cellular imaging [20,23,99].(b)*1,2,3-Triazolo[1,5-a]pyrazines*: For non-fused derivatives, the most common methods are: intramolecular cyclization of pyrazinyl hydrazones [36,42], formation of 1,2,3-triazolo[1,5-*a*]pyrazinium-5-olates from cyano and amide groups [37,40], or reaction of iodopropiolamides to form triazolopiperazine [43]. For benzo-fused derivatives (i.e., those containing quinoxaline or quinoxalinone), the most common methods are: cyclization of a ring-bound 1,2,3-triazole with an *ortho*-substituted amine [52] or nitro [45] group (if a nitro group, either PBu_3_ to give a quinoxaline [45] or FeCl_3_ [48] to give a quinoxalinone), cyclization of 1-azido-2-isocyanoarenes or 1-triazolyl-2-isocyanoarenes [54,55], or intramolecular cyclization of alkynes [53,57].*Primary Applications*: Primarily GABA_A_ modulating activity [3], and patents detailing use as Janus-family kinase modulators [101] or for the treatment of Huntington’s disease [100]. There also exist recent patents describing use as herbicides [102] and plant growth attenuators [103].(c)*1,2,3-Triazolo[4,5-d]pyridazines*: The most common synthetic method is reaction of a 4,5-dicarbonyl-1,2,3-triazole species with hydrazine to form the hydrazone, followed by acid or heat promoted cyclization [5,64,65,66,67,68,69,70,73,74,75,76,77,78,104]. The second most common method is treatment of the respective diaminopyridazine with nitrite [80,81,82,83]. Ring-opening/ring-closing of lactams has also been reported [84].*Primary Applications*: Use as highly conjugated linkers in triazole-based polymers [4] for the evaluation of solar cell materials is the main reported application.(d)*1,2,3-Triazolo[1,5-b]pyridazines*: The most common synthetic method is treatment of a keto-substituted pyridazine with *p*-bromophenyl hydrazine hydrochloride forming the hydrazone, then treatment with TBP in DCM [87,88]. A report of intramolecular cyclization of a diketo-oxime has been reported [86].*Primary Applications:* Benzo-fused or saturated piperazine-containing analogs are common. Notable reports include identification of a 1,2,3-triazole-fused quinoxalinone as inducing Rgg2/3-related gene expression in the human pathogen *Streptococcus pyrogenes*, as a potent DPP IV inhibitor [7], and as a BACE-1 inhibitor [6].(e)*1,2,3-Triazolo[4,5-c]pyridazines*: The most common method is cyclization of the respective diaminopyridazine with nitrite [3,74,91,92,93]. The intramolecular cyclization to form a pyridazine [94] or a tricyclic ylide have also been reported [22].*Primary Applications*: Outside of a patent [106] detailing immunoregulatory functions, no other applications exist in the literature at the time of this review.

The potential for new synthetic contributions is considerable for the triazole-fused pyrazines and pyridazines. Given the diversity of synthetic methods summarized in this review, new contributions that could be most beneficial are new routes to some of the precursors of the fused systems. In many of the reports cited, the starting materials are either not available commercially or are very expensive. For example, some diamino pyrazines are available as unsubstituted compounds or as halogenated derivatives, but all are USD 500–1000 per gram. Future studies of methods employing additional intramolecular cycloadditions leading to 1,2,3-triazolo[1,5-*a*]pyrazine derivatives would appear to have potential. Work on synthesis of the 1*H*-1,2,3-triazolo[4,5-*c*]pyridazines and the 1,2,3-triazolo[1,5-*b*]pyridazines would be welcome for these less frequently studied areas.

Overall, diverse methods exist for the preparation of 1,2,3-triazole-fused diazines, spanning the last seven decades with numerous reports in the last five years. Currently, drugs containing these ring systems remain scarce with only a handful of exceptions, particularly containing either the 1,2,3-triazolo[4,5-*b*]pyrazine or 1,2,3-triazolo[4,5-*c*]pyridazine scaffold. Applications of the aforementioned types of compounds span from medicinal chemistry into the development of dyes, probes, and inhibitors of enzymes implicated in various diseases. Despite this, there lies underrealized and exciting potential for employing triazolopyrazines and triazolopyridazines as diverse substrates in the generation of novel molecules with a wide array of applications.

## Data Availability

Not applicable.

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
