# Peer review of "Syntheses and Applications of 1,2,3-Triazole-Fused Pyrazines and Pyridazines"

_molecules, 2022, doi:10.3390/molecules27154681_

Round 1

Reviewer 1 Report

This paper is a comprehensive review of the published literature dealing with the synthesis of 1,2,3-triazole-fused pyrazines and pyridazines and related compounds. The manuscript is concise and easy to follow only the following remarks can be noted:

- Please include chemical yields in all schemes describing the synthesis of compounds of interest.

- In the introduction section, a general scheme summarizing the main methods for the synthesis of nuclei of interest would be desirable and very useful.

- In those schemes where the legend "commercially available precursors" appears (schemes 13, 26 and 29), it would be preferable to include chemical structures.

- It would be convenient to add a table with the different conditions used to obtain compound 45 (references 47 and 48).

- In line 404, it would be convenient to mention if the use of MnO2 was beneficial or not.

- Check the conditions on scheme 26 since there are no subsections a) and b).

- In figure 5 it would be convenient to include the compounds from reference 95 not just mention that similar structures were used.

Reviewer 2 Report

The manuscript entitled "Syntheses and Applications of 1,2,3-triazole-fused pyrazines and pyridazines" is a comprehensive summary of synthetic protocols for the preparation of heterocyclic systems bearing 1,2,3-triazoles fused with pyrazines and pyridazines in various combinations together with their further application in medicinal and materials chemistry.  The introduction part of the manuscript provides information on the content of the review, its general purpose and the literature area of performed investigation. A brief presentation of the types of heterocycle discussed in the manuscript, combined with selected examples of the practical application of each type, is provided in this part of the text.  The proper part of this review is distinguished into two parts, one dedicated to the introduction of the scope of synthetic procedures for all discussed types of heterocycles, and the second devoted to examples of the practical application of the aforementioned heterocyclic systems.  The final part is a well-prepared summary that helps the reader find the most important conclusions and general information.

In my opinion, the provided summary is highly instructive and useful for scientists involved in heterocyclic system development (and everyone interested in this field), especially for 1,2,3-triazole fused derivatives, and for that reason, the article deserves to be published in the Molecules Journal.

However, several editorial issues must be corrected before publishing.

- Keyword: 'application' is too general and does not provide information on the topic of the article.

- The compounds mentioned in lines 76-77 can be implemented in Scheme 1 or, like in other schemes, the frame 'commercially available precursors' can be used.

- 'N-' sometimes appears in italics, sometimes not (see lines: 99, 104, 181, 238, 240, 282, 288, 321, 322, 328)

- Line 192 'carboxamide N' and line 259 'quinoxalinone N' - I guess 'N' means nitrogen in combination with the previous issue is quite confusing – instead 'N' uses 'nitrogen' 

- The word 'via' should be written in italics (see lines: 123, 133, 135, 173, 242, 282, 310, 334, Scheme 27, 397, 529)

- Dot '.' put instead of space (see lines: 173, 184, 376, 390, 469)

- Line 143 - This sentence should be part of the description of Scheme 6 and not a previous paragraph.

- Scheme 6 - Explain the 'DIPEA' shortcut (like in other cases)

- Line 163 – Wentrup is mentioned as a single author of the research, but the next sentence starts from 'they' - should it be 'Wentrup and co-workers'? 

- The ranges of yields are recommended in brackets for lines 164 'in moderate yields' and 302 'excellent yields'. 

- Some unclear comments should be rewritten as 'amines need not be primary' (lines 246, 564), 'and contained within a ring' (line 252), 'Using a ring-closure method similar to that of Raghavendra and coworkers [40] and Cue and coworkers [44], Vogel and Lippmann [57] developed a route to derivatives of 66' (line 332), 'despite not having a second amine present' (line 419), 'pyridazines have since been found to be broader' (line 492)

- Scheme 26 - Letter 'c.' means concentrated H2SO4 (conc.)?

- Check and correct the reference part – half of the volume numbers are in italic and second not, some articles are provided with DOI, others not, '–' or '-' are used between pages randomly, 'et al.' should be in italic, check ref. 16, 33, 61, 95

In summary, the aforementioned article may be published in the Molecules Journal after minor revision including a careful review of the main text and reference parts.

Reviewer 3 Report

The present review article by Schoffstall and Hoffman summarizes the contemporary works on the synthesis of 1,2,3-triazole-fused pyrazines and pyridazines and their applications. This review article is focused on mainly two sections: Synthetic approaches, and applications. Recently, much effort has been paid for synthesis of 1,2,3-triazole-fused molecules and the authors provide a critical evaluation of the data available from these efforts in the form of a review article. Thus, this reviewer recommends publication of this article in Molecules. However, this reviewer has some comments and suggestions that the authors are encouraged to address

Comments:

1. As mentioned in the abstract, two typical synthetic routes for the construction of these ring systems are existent. The authors should provide a general scheme depicting these routes, highlighting the unique perceptions, in the introduction section of the manuscript.

2. Some compounds are not numbered (for example: compounds in Schemes 1, 4, 5, 7, 10 etc.). Please provide a unique number to all the compounds for their identification.

3.    One major thing I noticed in the manuscript is that it does not provide details about the reaction mechanism. The authors should discuss the reaction mechanisms for the novel reaction pathways, wherever applicable.

4.     In the conclusion section, besides drawing conclusions from the existing studies, it is desirable that the authors identify the gaps in the existing methodologies and point out the potential research areas to explore next.

Round 2

Reviewer 3 Report

The authors of the present manuscript "Syntheses and Applications of 1,2,3-Triazole-fused Pyrazines and Pyridazines" have successfully addressed all the comments raised by this reviewer. The manuscript is significantly improved to meet the standards of Molecules. Thus, this reviewer approves publication of this revised manuscript in Molecules